# Machine Learning-Based Analysis of Glioma Grades Reveals Co-Enrichment

**DOI:** 10.3390/cancers14041014

**Published:** 2022-02-17

**Authors:** Mateusz Garbulowski, Karolina Smolinska, Uğur Çabuk, Sara A. Yones, Ludovica Celli, Esma Nur Yaz, Fredrik Barrenäs, Klev Diamanti, Claes Wadelius, Jan Komorowski

**Affiliations:** 1Department of Cell and Molecular Biology, Uppsala University, 752 37 Uppsala, Sweden; karolina.smolinska@icm.uu.se (K.S.); ugur.cabuk@awi.de (U.Ç.); sara.younes@icm.uu.se (S.A.Y.); ludovica.celli@igm.cnr.it (L.C.); esma.yaz@std.medipol.edu.tr (E.N.Y.); fredrik.barrenas@icm.uu.se (F.B.); klev.diamanti@igp.uu.se (K.D.); 2Science for Life Laboratory, Department of Biochemistry and Biophysics, Stockholm University, 106 91 Solna, Sweden; 3Polar Terrestrial Environmental Systems, Alfred Wegener Institute Helmholtz Centre for Polar and Marine Research, 14473 Potsdam, Germany; 4Institute of Biochemistry and Biology, University of Potsdam, 14469 Potsdam, Germany; 5Institute of Molecular Genetics Luigi Luca Cavalli-Sforza, National Research Council, 27100 Pavia, Italy; 6Department of Biology and Biotechnology, University of Pavia, 27100 Pavia, Italy; 7Department of Biomedical Engineering and Bioinformatics, The Graduate School of Engineering and Natural Sciences, Istanbul Medipol University, Istanbul 34810, Turkey; 8Washington National Primate Research Center, Seattle, WA 98195, USA; 9Department of Immunology, Genetics and Pathology, Uppsala University, 751 85 Uppsala, Sweden; claes.wadelius@igp.uu.se; 10Swedish Collegium for Advanced Study, 752 38 Uppsala, Sweden; 11Institute of Computer Science, Polish Academy of Sciences, 01-248 Warsaw, Poland

**Keywords:** glioma, machine learning, batch effect, TCGA, co-enrichment, rough sets

## Abstract

**Simple Summary:**

Gliomas are heterogenous types of cancer, therefore the therapy should be personalized and targeted toward specific pathways. We developed a methodology that corrected strong batch effects from The Cancer Genome Atlas datasets and estimated glioma grade-specific co-enrichment mechanisms using machine learning. Our findings created hypotheses for annotations, e.g., pathways, that should be considered as therapeutic targets.

**Abstract:**

Gliomas develop and grow in the brain and central nervous system. Examining glioma grading processes is valuable for improving therapeutic challenges. One of the most extensive repositories storing transcriptomics data for gliomas is The Cancer Genome Atlas (TCGA). However, such big cohorts should be processed with caution and evaluated thoroughly as they can contain batch and other effects. Furthermore, biological mechanisms of cancer contain interactions among biomarkers. Thus, we applied an interpretable machine learning approach to discover such relationships. This type of transparent learning provides not only good predictability, but also reveals co-predictive mechanisms among features. In this study, we corrected the strong and confounded batch effect in the TCGA glioma data. We further used the corrected datasets to perform comprehensive machine learning analysis applied on single-sample gene set enrichment scores using collections from the Molecular Signature Database. Furthermore, using rule-based classifiers, we displayed networks of co-enrichment related to glioma grades. Moreover, we validated our results using the external glioma cohorts. We believe that utilizing corrected glioma cohorts from TCGA may improve the application and validation of any future studies. Finally, the co-enrichment and survival analysis provided detailed explanations for glioma progression and consequently, it should support the targeted treatment.

## 1. Introduction

Gliomas are heterogeneous brain and spinal cord tumors [1]. The expected survival of patients with glioma is extremely poor. In recent years, it was one of the leading cancer-related causes of death among most sex and age groups in adolescents and young adults [2]. The World Health Organization (WHO) in 2007 used cell types to classify gliomas into subtypes (astrocytoma, oligodendroglioma, oligoastrocytoma or ependymoma) or grades from I to IV [3]. In 2016, the subtyping system was updated according to molecular parameters such as the presence of a mutation in the *IDH1* gene [4]. However, the subtyping system update did not influence the grading system that is based on the histological criteria derived from a biological behavior of neoplasm. Specifically, WHO discerns four glioma grades that are defined as follows: grade I (GI or G1) with low proliferative potential, grade II (GII or G2) with low-level proliferative activity, grade III (GIII or G3) histological evidence of malignancy and grade IV (GIV or G4) cytologically malignant that is the most malignant form of glioma [5]. Here, the grading system was adopted from The Cancer Genome Atlas (TCGA) which classifies gliomas into lower-grade gliomas (LGG) including GII and GIII [6] and glioblastoma multiforme (GBM) including GIV.

Biological mechanisms behind any tumor progression, including glioma, are robust and affect many crucial signaling pathways. Numerous studies have identified alterations in the genome and characterized core pathways that are dysregulated. The study by [7] concluded that NF-κB participates in glioma angiogenesis that increases its malignancy. Interestingly, NF-κB is a critical factor that regulates immune response and the development of inflammatory diseases and cancer [8]. Furthermore, it has been established that the following pathways are disrupted in GBMs: (1) growth factor downstream signaling via phosphatidylinositol 3-kinase (PI3K) pathway; (2) apoptosis regulation via p53 signaling; (3) cell cycle regulation via cyclin-dependent kinases and retinoblastoma 1 signaling (RB1) pathway [9,10].

One of the most extensive data resources of transcriptomics datasets for gliomas is TCGA [11]. TCGA hosts a broad collection of samples sequenced with an RNA-seq, as well as other omics techniques. Recent studies have reported that various decision-unrelated sources of bias, i.e., batch effects, could occur among cohorts obtained from different sequencing facilities [12,13]. Importantly, a batch effect may influence the downstream analyses, especially when confounded with the outcome of the analysis, such as TCGA-LGG and TCGA-GBM. Furthermore, the impact of batch effect correction on datasets may remove biologically-relevant information that drastically affects the statistical analysis and thus, it shall be applied with great care [14]. To enhance reproducibility and limit variation, researchers created projects aiming to precompute and unify public cohorts such as recount2 or University of California, Santa Cruz (UCSC) Xena [15,16]. However, if the batch effect is highly confounded with an outcome of interest, a novel methodology needs to be designed and employed for correction.

In recent years, machine learning (ML) has been applied successfully in many scientific areas, including life sciences [17,18,19]. This popular field has been shown to effectively support knowledge mining and patterns recognition in big biological data [20]. For example, the prediction of a cancer outcome using ML techniques led to the detection of biomarkers and the exploration of novel ways of treatment [21]. An accurate prediction of the disease condition is a substantial challenge. However, the interpretation of ML models is also extremely challenging and increasingly fascinating [22]. Furthermore, the effect of batch on ML has also been studied [23]. The study showed that the bias is carried through the ML process and, thus it can affect the results and conclusions.

In this study, we corrected the strong batch effect between LGG and GBM cohorts from TCGA. For the corrected data, we provided a comprehensive ML analysis for two models of glioma grading: (1) GII vs. GIII and (2) LGG vs. GBM. To decrease variation from the batch effect, we used unified TCGA cohorts from the UCSC Xena repository [16] that were recomputed under the UCSC Toil RNA-seq pipeline. The analysis was divided into two stages. First, we focused on analyzing differentially expressed genes (DEGs). Second, we performed a single-sample gene set enrichment analysis (ssGSEA) that was followed by a comprehensive ML evaluation and analysis. We applied ssGSEA to detect the most accurate Molecular Signatures Database (MSigDB) [24,25,26] collections that discern between glioma grades. Next, topmost collections were used to determine dependencies among annotations that revealed co-enrichment for glioma grading. Finally, we validated our results and evaluated the survival of the glioma patients for the most co-enriched annotations.

## 2. Materials and Methods

This study focused on developing a methodology for ML-based analysis of glioma cohorts from TCGA (Figure 1). First, we aimed at correcting the strong batch effect that biased the ML analysis performance. After successfully correcting the datasets, we performed two separate analyses: (1) Differential gene expression analysis; (2) ML analysis on ssGSEA scores based on MSigDB collections. Finally, two cohorts from the Chinese Glioma Genome Atlas (CGGA) [27] were used for validating the results. Below, we included detailed descriptions of the applied methods.

### 2.1. Preprocessing of Gene Expression Datasets

Initially, we analyzed raw transcriptomics data from the TCGA repository. To perform the analysis, we collected RNA-seq datasets from Genomic Data Commons (GDC; [28], including TCGA-LGG and TCGA-GBM cohorts. Using principal component analysis (PCA), we observed a strong variability between GDC-based cohorts (Appendix A). Therefore, we used a unified and transcript per million-normalized dataset from UCSC Xena Toil RNA-seq recompute data hub, which contains merged cohorts TCGA, TARGET and GTEx [29]. We chose this cohort to allow future expansion of the analysis as it also includes TARGET and GTEx. However, even for the unified cohort, strong variability was still visible (Appendix A). Therefore, we attempted to correct for unknown sources of batch effects (Appendix A) [30]. Using an ML evaluation and Student’s *t*-test, we examined how bias influences classification between particular grades groups (Appendix A). We found that classifiers result in very high quality for randomly chosen genes (Appendix A) and an unusually high fraction of these genes being DEGs (Appendix A).

In the next step, we examined principal components (PCs) of the unified cohort and we retrieved Gaussian mixtures (GMs) from the first PC (PC1) (Appendix A). This allowed us to detect mixtures that corresponded to hidden groups of samples. Based on PC1, two GMs could be distinguished (Appendix A). These are GM1 that contained LGG and GBM samples, and GM2 that contained the vast majority of LGG samples. Therefore, we divided the unified TCGA dataset according to GMs (Appendix A). Specifically, the final GM1-based dataset contained 151 GBM and 108 LGG samples, and the GM2-based dataset collected 231 GII and 168 GIII samples (Table 1). We also evaluated which samples were separated based on GMs. Here, we provided global PCA and local t-SNE approaches (Appendix A). To show the potential source of the batch effect, we visualized tissue source sites (TSSs) for all samples (Appendix A).

We first evaluated the GM1-based dataset and observed that classifiers built with randomly selected genes are closer to the accuracy of permutation tests (Appendix A). In addition, as expected the fraction of DEGs decreased (Appendix A). Next, we ran surrogate batch effect analysis on GM1-based samples using two methods, namely “leek” and “be” (Appendix A). Furthermore, after applying batch effect correction, we selected protein-coding genes (Appendix A) using a reference file from HUGO Gene Nomenclature Committee [31]. As a result, datasets contained 19,028 protein-coding genes. To evaluate if the biological information was not affected by batch effect correction, we again performed the *t*-test on the GM1-based dataset that resulted in 255 significant genes with *p* value adjusted for false discovery rate (FDR) less than 0.001. Next, GM2-based data were processed in a similar fashion. We examined the variability between GII and GIII (Appendix A), where no batch effect was visible and the correction step was omitted. Finally, we performed a *t*-test between GII and GIII and found 439 significant genes (FDR-adjusted *p*-value < 0.001). For the list of significant genes, we run functional profiling with *gProfiler*, using all available databases, that revealed sets of significant (FDR-adjusted *p*-value < 0.05) pathways. In addition, we checked the influence of sex and age on GM1- and GM2-based datasets (Appendix A).

### 2.2. DNA Methylation Data

The DNA methylation data were used to examine epigenetic changes in DEGs for samples corresponding to GM1 and GM2. The DNA methylation profiling was based on the Illumina Infinium HumanMethylation450 platform for 685 samples corresponding to samples in transcriptomics analysis. The dataset (GBM-LGG) was downloaded from the UCSC Xena browser. CpG sites with no recorded beta values were filtered out prior to downstream processing. After filtering, 364,859 CpG sites remained. To check for possible batch effects, we visualized the dataset using PCA. We annotated CpG sites with their associated genes (Appendix A). The average beta values across different groups were compared using a non-parametric Wilcoxon test.

### 2.3. ssGSEA Analysis

To further analyze GM1- and GM2-based datasets, we employed ssGSEA, a single-sample extension of GSEA. Using the ssGSEA approach, we transformed all the variables from gene expression values to the degree of enrichment. The single-sample approach decreases the variability of datasets suffering from confounding factors. To perform ssGSEA, we used a method proposed by [32] (Appendix A). We first selected all 20 collections from MSigDB v7.4 and then ran ssGSEA on glioma cohorts. These collections included the following gene sets: hallmark, positional (PG), chemical and genetic perturbations (CGP), BioCarta, Kyoto Encyclopedia of Genes and Genomes (KEGG), Pathway Interaction Database (PID), Reactome, WikiPathways (WP), microRNA targets (MIR), transcription factor targets (TFT), cancer gene neighborhoods (CGN), cancer modules (CM), gene ontology cellular component, biological process and molecular function (GOCC, GOBP and GOMF, respectively), human phenotype ontology (HPO), oncogenic signatures (Onco), ImmuneSigDB (Immuno), vaccine response gene sets (VAX) and cell type (CT).

Some ML methods, such as rule-based learning, require discrete variables to perform learning. In ssGSEA, enrichment scores describe the activity degree of a given gene set. Each score represents the enrichment degree to which the genes are simultaneously down- or up-regulated for a single sample. The ssGSEA scores were discretized with equal frequency, as it was done in rule-based modeling, into three levels of the degree: low, medium and high. For instance, a high ssGSEA degree means that there are many down- or up-regulated genes from the given gene set in a particular glioma sample. We believe that such simplification of the ssGSEA degree could lead to improved interpretability.

### 2.4. ML Evaluation

To evaluate the MSigDB collections, we performed an ML analysis before and after applying Monte Carlo feature selection (MCFS) [33] that is a decision tree-based non-linear method. According to the ML evaluation (Appendix A), we selected the top three most accurate collections for discerning grades. In order to evaluate the classification abilities of collections, we selected five different and well-established ML approaches [34], namely: sequential minimal optimization (SMO) for training support vector classifiers [35], instance-based learning algorithms (IBk) that is an extension of *k*-nearest neighbors algorithm [36], bagging predictors [37], J48 that generates C4.5-based decision trees [38] and repeated incremental pruning to produce error reduction (JRip) that creates classifiers by rule learning algorithm [39]. We based our choice criterion on mixing several well-known black-box and interpretable ML methods. As the datasets included an uneven distribution of decision classes, we applied undersampling of the majority class to match the size of the minority class. For instance, the number of balanced classes in the case of GII vs. GIII was equal to the total number of GIII samples, i.e., 168 samples (Table 1). The undersampling was performed 20-times in order to obtain balanced datasets. Next, the ML modeling was performed with 10-fold cross-validation (CV). In addition, we performed a permutation test for each model. The permutation test has been performed by randomly shuffling the decision classes. The test was included within an undersampling loop and performed with 10-fold CV.

We employed two well-known classification quality measures in this work, namely accuracy (ACC) and area under the ROC curve (AUC). The ACC was used for the ML evaluation of datasets before applying ssGSEA analysis. After applying ssGSEA, we used the AUC measure for evaluation. We used undersampling in all experiments, we believe these metrics can be used interchangeably.

### 2.5. Interpretable ML

To find dependencies between annotations and provide interpretable classifiers, we generated rule-based models (RBMs) with *R.ROSETTA* [40]. The method uses a rough sets theory for producing a set of IF-THEN rules that constitute an RBM [41]. The set of rules was initially created using a Boolean reasoning approach. However, since a Boolean reasoning approach is a non-deterministic polynomial hard problem, several algorithms called reducers have been developed to tackle this dilemma. Here, we used the Johnson reducer method that produces a high fraction of significant rules and does not overestimate their total amount [40]. Importantly, we have created RBMs only for the topmost MSigDB collections selected based on the AUC value. The rules were further filtered according to their *p*-value (FDR-adjusted *p*-value < 0.01). Notably, such rules are directly interpretable and reflect co-predictive mechanisms among features. As in the case of well-known co-expression analysis, such dependencies may reflect biological interactions. However, rules characterize non-linear, local and supervised dependencies of features. As in the case of previous ML evaluations, we applied undersampling and 10-fold CV for obtaining RBMs. Importantly, equal frequency discretization of ssGSEA scores was performed within the CV loop.

### 2.6. Rule-Based Networks of Co-Enrichment

Here, we presented co-predictive mechanisms as a co-enrichment that is defined as two or more annotations being simultaneously enriched for a specific group of samples regarding the decision class, i.e., glioma grade. Usually, annotations are treated independently, but we assumed that annotations might have complementary functions [42]. Such annotation-annotation dependencies have been successfully investigated for evaluating drug effects [43]. In general, co-enrichment has been shown as an interesting concept for analyzing data in the form of a network [44]. Thus, we are aware of its high importance in analyzing complex diseases such as glioma.

RBMs were further visualized using a rule-based network approach with *VisuNet* [45]. This approach transforms a set of rules into a network. Here, the network represents annotations and their values as nodes and rule-derived connections as edges. We used the decision coverage value to define the size of nodes. Furthermore, the feature enrichment score degree corresponds to the color of nodes and the adjusted connection strength between two nodes from a rule defines the color and width of edges. We adjusted connection values on the network to normalize the co-enrichment that may occur due to the overlapping gene sets. To obtain normalized connection values on networks and total correlation values [46](Appendix A) on heatmaps, we used the following formula:(1)$$v_{norm}=v*\left(1-\alpha \right),$$
where v is the connection value between two nodes or total correlation value and α is the degree of overlapping genes between two gene sets. For instance, if there are no overlapping genes between gene sets, then α=0 and $v_{norm}=v$. All networks presented in the paper were created for the 20 most connected nodes of the top 10% rules ranked by the connection value of rules. Finally, as another level of visualization, we used a concept of arc diagrams for displaying particular nodes, i.e., nodes of interest (NOI), from networks.

## 3. Results

### 3.1. Data Correction

As a result of comprehensive data preprocessing, we detected two subsets of samples within TCGA glioma cohorts. We assumed that these subsets correspond to the hidden, i.e., unknown, batch effect. However, we suspect that this batch effect is related to TSS (Appendix A) as it is clearly visible that the majority of GBM-related TSSs are visualized as a separate cluster. In other words, source sites (hospitals, universities, etc.) are highly confounded with the decision class LGG vs. GBM. Thus, we further used the surrogate variable analysis on the GM-based subset that assisted in removing the batch effect from LGG vs. GBM data (Appendix A). Therefore, we enclosed a table (Appendix A) that may help in future studies of TCGA glioma datasets for more accurate analysis, which includes TCGA sample IDs, GM groups and grade information. We believe that GM modeling, together with PCA, can be applied in similar situations to correct highly confounded batch effects.

### 3.2. DEGs Evaluation

First, we identified lists of highly significant DEGs (FDR-adjusted *p*-value < 0.001) for GII vs. GIII (Appendix A) and LGG vs. GBM (Appendix A) that we used to perform functional profiling (Appendix A). Based on the results, we examined the most significant and interesting functional annotations for discerning glioma grades. For GII vs. GIII (Appendix A), we observed that the cell cycle, p53, DNA replication and Fanconi anemia were among the most significant pathways of KEGG. In addition, from GOBP, we noticed that the cell cycle is highly enriched. Furthermore, GOCC suggested that the list of DEGs is highly enriched for chromosome-related annotations. Interestingly, Reactome and WP also pointed towards cell cycle-related annotations. In addition, several cancer-related pathways from WP were enriched.

For LGG vs. GBM (Appendix A), we observed that nonsense-mediated decay (NMD) was highly significant in the Reactome database. This finding corroborates a recent discovery that showed modulation of NMD promoting the growth of GBM in humans [47]. Based on the Reactome, the metabolism of RNA is significantly important for GBM grading processes. In addition, rRNA and mRNA processing signaling pathways were significantly enriched. All significant GOMF annotations pointed towards binding processes, e.g., RNA or nucleic acids binding. In both grading-related cases, no brain-related tissues were detected among human protein atlas annotations, highlighting no tissue-specific DEGs. In the next step, using CGGA data, we validated sets of DEGs. For validation, we used preprocessed and normalized RNA-seq CGGA datasets. While using CGGA, we assumed that LGGs are samples marked as GII or GIII, as well as it was done by TCGA. Here, we examined how many TCGA-based DEGs were in two CGGA cohorts (batch 1 and 2) (Table 2). We observed a good overlap of DEGs between these independent two cohorts.

Finally, we intersected both gene lists, i.e., GII vs. GIII and LGG vs. GBM, and found 6 DEGs in common: *IGIP*, *NSMCE2*, *CNIH4*, *NONO*, *CKLF* and *RAN* (Appendix A). We then validated the expression of these genes using CGGA batch 1 and batch 2. Both batches confirmed that these 6 DEGs were differentially expressed in the validation sets for discerning GII vs. GIII and LGG vs. GBM (Appendix A). In addition, we examined DNA methylation data corresponding to GM1- and GM2-based samples (Appendix A). We further checked DNA methylation profiles of the 6 shared DEGs (Appendix A). We found several differentially methylated regions (DMRs). Interestingly, there were more DMRs in 6 common DEGs for LGG vs. GBM, which may suggest that more robust epigenetic changes are visible while progressing to a higher grade.

### 3.3. ML for ssGSEA

We evaluated MSigDB collections using an ML approach (Figure 2). We applied learning to each collection separately to choose the best collections discerning glioma grades. To further assess the MSigDB collections, we performed feature selection on each collection and selected a balanced number of important features (Appendix A). Afterward, we selected the three top collections for GII vs. GII CGP, BioCarta and PID, and for LGG vs. GBM GOCC, GOBP and WP.

Interestingly, all three selected collections for GII vs. GIII were curated gene sets, while two out of three collections for LGG vs. GBM were ontology gene sets. This may suggest that more biological processes on the cellular structure level are disrupted for progression to a higher glioma grade. Furthermore, we observed that cancer-related collections were also highly predictive in both cases (Figure 2 and Figure 3). As expected, differences between LGG and GBM are more prominent than between GII and GIII. In addition, feature selection improved the quality of all models. Thanks to feature selection, we received fewer features that in turn enhanced the interpretability of the models. In addition, MCFS application was necessary to balance the number of features across compared collections (Appendix A). From MCFS, we presented the most important annotations discerning between grades. For instance, “Fanconi”, “cell cycle” and “Spermatocyte” [48] were annotations with the highest relative importance (RI) values discerning GII from GIII.

To create rule-based models, we used the R.ROSETTA method that resulted in highly accurate classifiers (Appendix A). The best collection for discerning GII from GIII was CGP, which resulted in 0.71 AUC, and the best collection for discerning LGG from GBM was GOCC, which resulted in 0.84 AUC. For each of the two comparisons, we created a joint rule-based model by merging the features from the top three collections. However, AUCs for joint models were not better than the best AUC of single models (Appendix A). Finally, we validated rule-based models by classifying CGGA batch 1 and batch 2 gene expression datasets. For the validation, we used TCGA-derived annotations that classified grades with similar performance (Appendix A). In addition, we estimated total correlation values among all annotation pairs for the topmost collections. We calculated this correlation for ssGSEA scores discretized into three levels with equal frequency binning.

### 3.4. Glioma Co-Enrichment

We visualized RBMs as networks to evaluate co-enrichment mechanisms among annotations for topmost MSigDB collections (Figure 4, Appendix A). As networks, displayed for the entire set of rules, were unbalanced with respect to decision classes, we generated balanced networks for each decision class separately (Figure 4, Appendix A). We discovered co-enrichment mechanisms among the most significant rules (*p*-value < 0.01). We included sets of significant rules in Appendix A, for the networks Figure 4A,B, respectively. Notably, these mechanisms are local and correspond to a group of patients, i.e., the rule is supported by a set of samples that fulfill its conditions. Connections values on all networks were normalized according to the number of genes shared between gene sets.

By analyzing networks and investigating NOIs, several findings can be described. Here, the main hub in the GII and GIII networks (Figure 4A,B) was “amplification hot spot 15” [49]. In both subnetworks (Figure 4A,B), this node is connected to several annotations, among others “Wilms tumor vs. fetal kidney 2 up” [50] and “Soft tissue tumors PCA2 up” [51]. The latter indicated that these pathways may be linked to activating the human set of specific oncogenes during progression from a lower to a higher grade. We could also observe more generic annotations, for instance, “cell cycle” or “G2 phase” (Figure 3 and Appendix A). Here, the “G2 phase”-related gene set is more relevant than “G1 phase” for GIII (Figure 3E). However, several authors have reported changes in cell cycle regulation under various circumstances [9,52,53,54]. Interestingly, a high degree of enrichment for the “Aurora A” pathway played a major role in grading (Figure 3G and Appendix A). The “Mitotic Aurora A kinase” (AurA) pathway is an essential factor in the survival, radio-resistance, self-renewal and proliferation of glioblastoma cells [55]. Its potential therapeutic abilities have also been investigated [56]. Here, we found that the enrichment of the AurA pathway differed between GII and GIII. More specifically, we found that “Aurora A” was a highly co-enriched pathway for GII vs. GIII (Appendix A). Interestingly, “Aurora B” appeared to be highly interactive in validation cohorts (Appendix A). Importantly, both Aurora pathways were investigated for treating cancer [57]. Furthermore, we found that the Fanconi anemia pathway played a crucial role in glioma grading (Figure 3G, Appendix A). Recently, its potential therapeutic role has been described [58]. From other topmost MSigDB collections discerning between GII vs. GIII, we found evidence of a local co-enrichment between “P53” and “PDGF” pathways, i.e., strong connection in a network (Appendix A). It is well-known that alterations in the *P53* gene promote tumor development, malignancy and resistance to radio and drug therapy [59,60]. Furthermore, the *PDGF* gene is one of several growth factors participating in glioma angiogenesis [61]. Here, we provided a hypothesis that these two pathways may be co-dependent.

Next, we investigated results obtained for grading to GBM (Figure 4C,D, Appendix A) that were further validated (Appendix A). Intermediate density lipoprotein (IDL) was NOI in LGG and GBM networks (Figure 4C,D). This annotation was also highly co-correlated in both validations sets (Appendix A). A high connection between IDL and microtubule-related annotations was visible in all cases. The role of microtubules in the degradation of lipoproteins has been previously identified [62] and this result might provide additional evidence of its potential role as a therapeutic target for GBM treatment. Chylomicron and IDL particle detected from GOCC collection (Figure 3D, Figure 4C,D and Appendix A) and sterol transport-related annotations from GOBP (Figure 3F, Appendix A) may suggest a link to cholesterol-related mechanisms and glioma grading. Recent studies have elaborated that cholesterol metabolism may be a potential therapeutic target in glioma [63,64]. In this study, we found that the degree of enrichment of cholesterol-related pathways strongly differs between LGG and GBM. Interestingly, we found that cholesterol-related pathways were highly co-enriched and may affect microtubule organization in the case of GBM (Appendix A). Thus, annotations related to cholesterol could be further investigated for their therapeutic potential in GBM patients. Moreover, we found a high activation degree of “methionine de novo and salvage” pathway (Figure 3H and Appendix A). The survival and proliferation of cancer cells were shown to be dependent on methionine levels [65]. Finally, we found that microtubule organization differed between LGG and GBM (Figure 3D and Figure 4C,D). The inhibition of microtubule dynamics was explored previously for its potential in GBM treatment [66].

### 3.5. Survival Analysis

We performed a survival analysis [67,68] for selected NOIs of the RBM networks (Figure 4, Appendix A). Here, we obtained overall survival and its status from cBioPortal [69,70]. The survival analysis was done to determine the degree of a given annotation enrichment corresponding to discrete levels obtained with equal frequency discretization. The high degree of enrichment for both NOIs from CGP-based networks, i.e., “Amplification hot spot 15” and “Hypoxia not via KDM3A”, contributed to poor survival for LGGs (Figure 5A,B). We observed that a low degree of enrichment, i.e., low activity, for “IDL particle”, “negative regulation of sterol transport” and “Valproic acid” corresponded to poor survival in samples with higher glioma grade. Interestingly, “Valproic acid” (Figure 5C,G,K) has been recently proposed as a promising therapeutic target for gliomas [71]. We also noticed that microtubule-related annotations did not influence the overall survival (Figure 5D,H). For LGGs, we observed that a high degree of enrichment with “G2 phase” (Figure 5F) played a role in overall survival, while the impact of the “G1 phase” was insignificant (Figure 5E). Furthermore, a high degree of AurA and P38/MK2 enrichment was significantly associated with poor survival in LGGs (Figure 5I,J).

## 4. Discussion

This study provides hypotheses of co-enrichment between glioma grade-related annotations regarding their high predictivity. There are several advantages of discovering co-enrichment between annotations. From the study by [72], we know that interaction may occur between two perturbed pathways that can lead either to increased or decreased disease risk. Several studies [73,74,75] have shown that discovering an interaction among pathways may improve therapy for treating cancer. Thus, we believe that the findings of this analysis may provide insights for future research and aid in novel ways of clinical treatment.

It is imperative to provide non-biased analyses in bioinformatics. Thus, we aimed at comprehensive preprocessing to perform unbiased ML analysis and retrieve biologically meaningful results in this work. As we aimed at performing ML analysis, we focused on the global structure of data, i.e., PCA, in order to investigate a low number of clusters enriched with a high number of samples. We showed that the local structure of the data, i.e., t-SNE analysis, generated very similar clusters (Appendix A). Thus, we concluded that in this study, PCA and t-SNE approaches were comparable. Moreover, we provided a thorough benchmarking of several ML methods that revealed specific MSigDB collections corresponding to glioma that may guide future research. However, the predictivity of specific collections can be disease-specific, so it shall be estimated separately while analyzing other types of cancers or other diseases.

In this work, we transformed gene expression into annotations, as we believe that pathways are more universal than genes. As biology is robust and diverse, it is more reasonable to perform analysis based on pathways. For instance, assuming that expression alterations of Gene A and Gene B lead to a change in Pathway X. In such case, Patient 1 that has a change in expression of Gene A and Patient 2 that has a change in expression of Gene B would be merged into a common group of patients that have a change in Pathway X. Thus, the variability is decreased and a common unit, i.e., a pathway, is established. There are several advantages [76] of applying pathway enrichment methods, such as aggregating information, reducing data dimensionality, enhancing the interpretation of results, identifying drug targets, providing better comparability within the same omics technology or between various omics technologies. Despite the wide range of advantages, there are also limitations [76,77]. Among others, effectiveness is determined by the strength of signals from multiple genes, databases are biased toward well-studied genes and pathways, and interactions among genes are neglected, i.e., gene independence is assumed. Moreover, here we used ssGSEA and thus, we are limiting the data space to a single-sample approach. The reason for that was to follow the idea proposed by [78] to provide an additional layer that prevented the strong batch effect that occurred for this particular dataset. Furthermore, we used data on a discrete scale as it is necessary for our rule-based approach [79]. Thanks to this, we improved the interpretability of results and reduced the influence of noise in the data. However, the discretization process reduced the information and neglected the continuous nature of the data. Lastly, we performed the basic survival analysis for relevant annotations. To explore various survival tasks in a more comprehensive way, we encourage using a more advanced and recent approach such as DeepPAM [80,81].

We observed that most of the co-enrichment mechanisms detected via networks (Figure 4, Appendix A) for LGG vs. GBM were also observed in validation analyses of the CGGA batches (Appendix A). In contrast, fewer co-enrichment mechanisms describing GII vs. GIII networks could be validated with the CGGA batches. Thus, it is highly possible that in the case of GII vs. GIII, ML models with low accuracy could have affected the analysis and disrupted the validation. In general, we could observe that the quality for discerning GII and GIII is not very high. On the other hand, the quality of interpretable rule-based models is above median compared to other ML techniques (cf. Figure 3A,B and Appendix A). Thus, we provided evidence that interpretable learning is not only legible but also produces high-quality models [82]. Importantly, this work demonstrates the results of non-linear dependencies of features. Thus, it may be also interesting to investigate linear dependencies of features in the future. For instance, by using other approaches such as importance-based sequential procedure [83].

We provided a simple normalization method to avoid false positives due to overlapping gene sets for co-enrichment. While normalizing networks or validating heatmaps, using Equation (1), we observed that overlaps of genes between gene sets are minor. This may be due to comparing gene sets within separate MSigDB collections. Thus, the normalization adjusted the final results slightly. Here, validation represented global co-enrichment with respect to decision classes, while networks showed local co-enrichment. Thus, the validation of co-enrichment mechanisms provided a general overview of our findings.

Taken together, this study provided a methodology that not only demonstrates how to perform batch effect removal towards ML analysis but also reveals potential interactions among pathways using an interpretable ML approach. We supported the analysis with statistical measures and tests. We believe that our findings can serve as potential therapeutic targets that could improve glioma treatment on various grade levels. The major future perspective is that these hypotheses can be validated experimentally to ensure our findings and incorporate them into glioma treatment.

## 5. Conclusions

A key challenge in bioinformatics is to perform the analysis in an unbiased, repetitive and accurate way. This study demonstrated how to remove a strong batch effect from TCGA glioma datasets and perform comprehensive ML analysis. Herein, LGG and GBM cohorts included a strong batch effect confounded with outcome classes. In such cases, it is essential to correct the batch effect, but it has to be done carefully in order to keep the biological information included in the data. Furthermore, this work describes co-enrichment mechanisms that reflect robust processes for glioma progression. Notably, the proposed methodology is generic and can be used on any problematic data. To the best of our knowledge, this is the first co-enrichment analysis of glioma grades using rule-based learning.

## Figures and Tables

**Figure 1 cancers-14-01014-f001:**
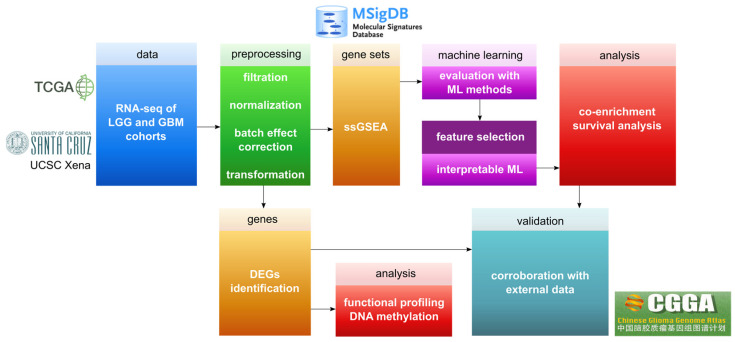
Overview of the pipeline applied in this work. Following preprocessing were two separate analyses: The lower tier of the pipeline illustrates the steps employed for identification and basic analysis of DEGs, while the upper tier demonstrates the steps for ssGSEA analysis based on ML approaches using MSigDB collections. The final step illustrates the validation of the results.

**Figure 2 cancers-14-01014-f002:**
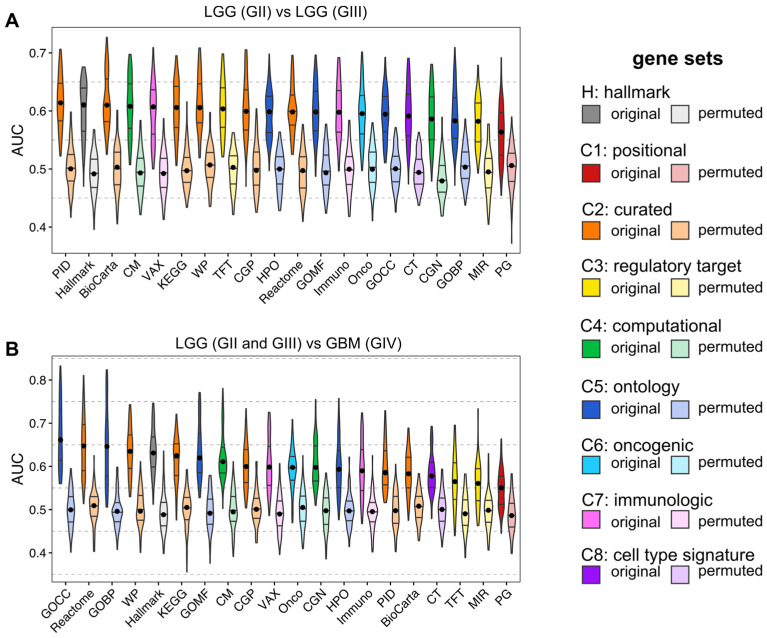
Evaluation of MSigDB collections using ML models for discerning glioma grades using ssGSEA scores for classifying (**A**) GII vs. GIII and (**B**) LGG vs. GBM. Five different ML approaches were used: SMO, IBk, Bagging, J48 and JRip. Each ML method was undersampled 20-times with 10-fold CV. The median was marked with a black dot on each violin.

**Figure 3 cancers-14-01014-f003:**
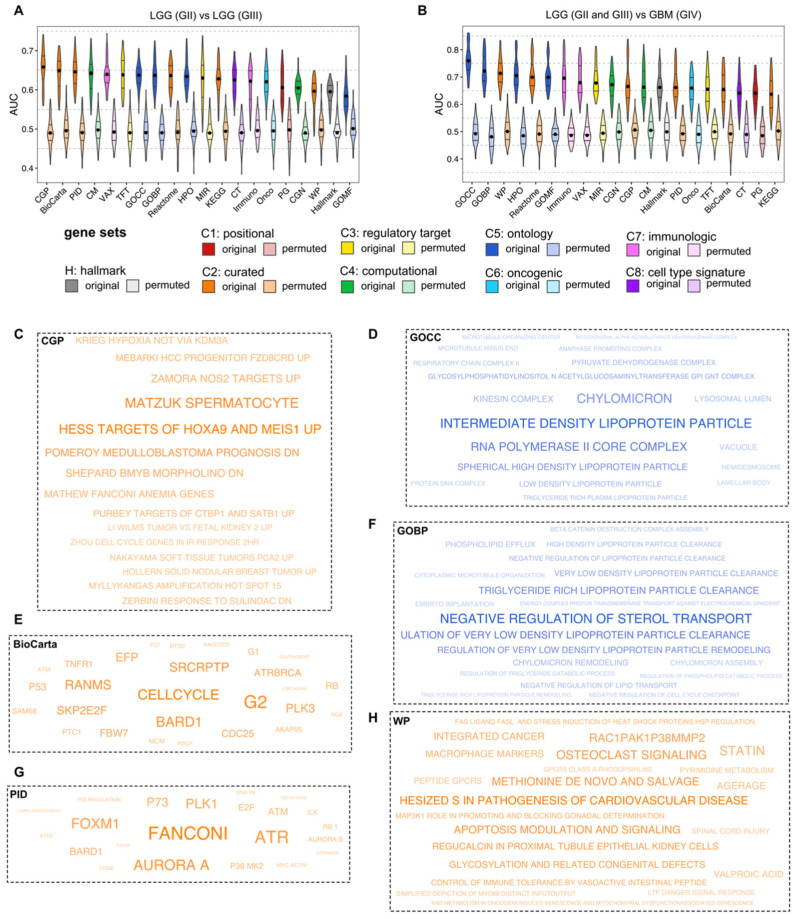
(**A**,**B**) Evaluation of MSigDB collections using ML models with feature selection for discerning glioma grades using ssGSEA scores. Five different ML approaches were used: SMO, IBk, Bagging, J48 and JRip. Each ML method was undersampled 20-times with 10-fold CV. The median was marked with a black dot on each violin. Panels (**C**,**E**,**G**) represent MCFS results for the top three collections for GII vs. GIII. Size of annotations represents RI values from MCFS. Panels (**D**,**F**,**H**) represent MCFS results for the top three collections for GII vs. GIII. Size of annotations represents RI values from MCFS.

**Figure 4 cancers-14-01014-f004:**
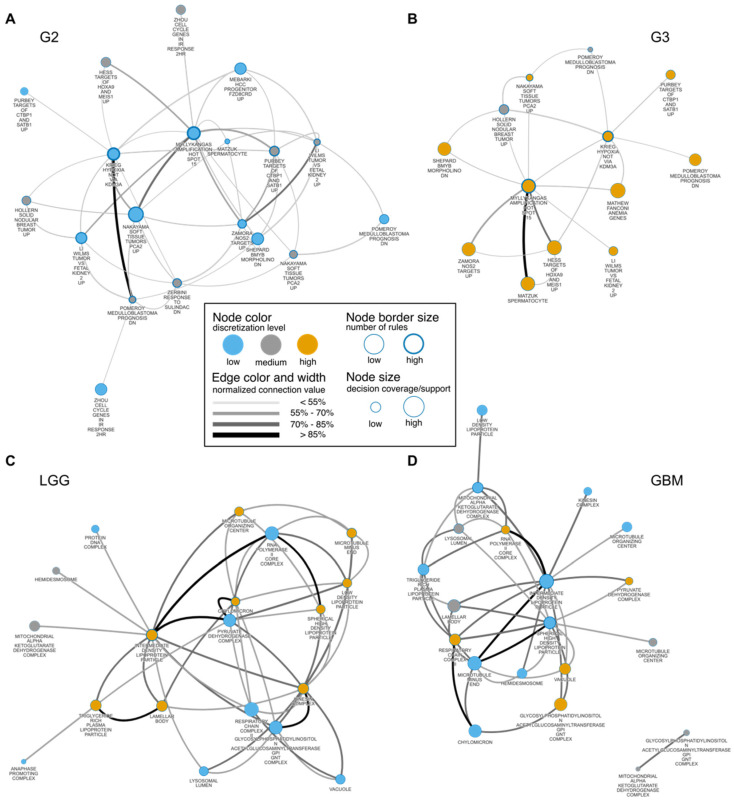
Rule-based network displaying the most relevant co-enrichments of annotations obtained from (**A**,**B**) the CGP collection for the GII vs. GIII model (Appendix A) and from (**C**,**D**) the GOCC collection for the LGG vs. GBM model (Appendix A). The networks show the 20 most connected nodes obtained from the top 10% of significant rules (FDR-adjusted *p*-value < 0.01) based on the rule connection. Connection values of nodes and edges represent a strength of co-enrichment from the classifier. Subnetworks were generated separately with respect to the decision class for each RBM.

**Figure 5 cancers-14-01014-f005:**
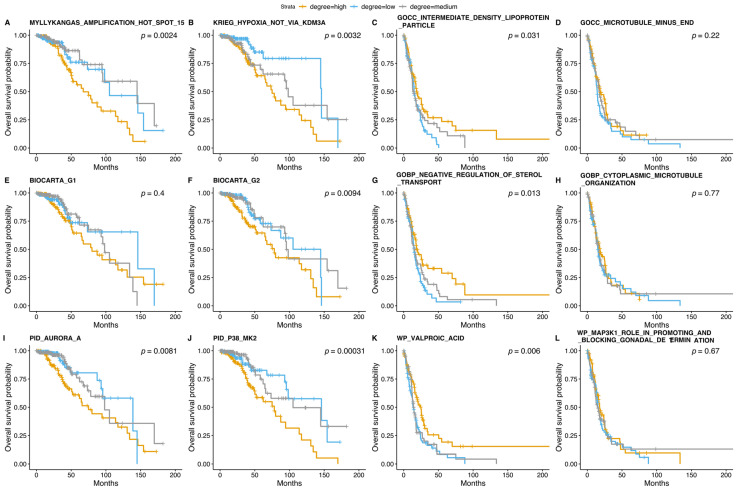
Survival curves of several NOIs characterized based on rule networks. We investigated NOIs for the topmost predictive three MSigDB collections discerning glioma grades: (**A**–**D**) CGP and GOCC, (**E**–**H**) BioCarta and GOBP, and (**I**–**L**) PID and WP. Each plot displays a *p*-value that was estimated with the default set of parameters while constructing the curves (Appendix A).

**Table 1 cancers-14-01014-t001:** A summary of sample amounts used in the analysis to obtain and validate results. In total, 1671 publicly available samples were used in this analysis.

TCGA				CGGA					
GM1		GM2		Batch 1			Batch 2		
GII	GIII	LGG	GBM	GII	GIII	GBM	GII	GIII	GBM
231	168	108	151	188	255	249	103	79	139

**Table 2 cancers-14-01014-t002:** The results of DEG list validation with CGGA cohorts. *p* values in validation cohorts were FDR-adjusted.

CGGA Batch 1				CGGA Batch 2			
GII vs. GIII		LGG vs. GBM		GII vs. GIII		LGG vs. GBM	
*p* < 0.001	*p* < 0.05	*p* < 0.001	*p* < 0.05	*p* < 0.001	*p* < 0.05	*p* < 0.001	*p* < 0.05
62%	88%	27%	44%	85%	96%	52%	71%

## Data Availability

Data used in this study were acquired from public resources: https://xenabrowser.net/hub/ (accessed on 6 April 2021), http://www.cgga.org.cn/download.jsp (accessed on 10 February 2021) and http://www.gsea-msigdb.org/gsea/msigdb/ (accessed on 27 April 2021).

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
