# Peer review of "Machine Learning-Based Analysis of Glioma Grades Reveals Co-Enrichment"

_cancers, 2022, doi:10.3390/cancers14041014_

Round 1
Reviewer 1 Report
The paper presents a methodology to correct strong batch effects from The Cancer genome Atlas dataset and use machine learning to estimate glioma grade-specific co enrichment mechanisms. Authors performed unbiased ML analysis using neural networks, kNN, rule-based classifier, decision trees, and bagging. They study could be utilized for reducing bias and less noisy data needed for ML techniques. There are few comments as follows:
- Authors have developed a methodology to correct batch effect, but it is not explained what batch effect is.
- It is not clear what classifier is used for the machine learning in the analysis for Figure S2. Machine learning is a very general and board term. It is needed to be more specific about the technique. Also, what is the unit of y-axis in Figure S2(A-D). If it is percentage, needs to be added to the y-axis.
- Authors performed under-sampling to balance data for 20-times. But what was the size of data after under-sampling?
- References needs to be provided for classifiers in section 2.4. Also, what are SMO, IBk, J48, JRip in section 2.4? are they specific types of classifiers? They should be spelled out with providing a reference.
Author Response
Response to Reviewer 1 Comments
The paper presents a methodology to correct strong batch effects from The Cancer genome Atlas dataset and use machine learning to estimate glioma grade-specific co enrichment mechanisms. Authors performed unbiased ML analysis using neural networks, kNN, rule-based classifier, decision trees, and bagging. They study could be utilized for reducing bias and less noisy data needed for ML techniques. There are few comments as follows:
Authors have developed a methodology to correct batch effect, but it is not explained what batch effect is.
We thank the reviewer for this important comment. We defined what batch effect is in the introduction. We also clarified what kind of batch effect it potentially is in section 3.1 “Data correction”.
It is not clear what classifier is used for the machine learning in the analysis for Figure S2. Machine learning is a very general and board term. It is needed to be more specific about the technique. Also, what is the unit of y-axis in Figure S2(A-D). If it is percentage, needs to be added to the y-axis.
We specified the algorithms used in ML evaluations and added descriptions to Figures S2, S5 and S6. We also added a unit to the y-axis of ACC values in Figures S2, S5 and S6.
Authors performed under-sampling to balance data for 20-times. But what was the size of data after under-sampling?
This information is given in chapter 2.4. To clarify this issue, we provided an example for GII vs GIII and referred to Table 1.
References needs to be provided for classifiers in section 2.4. Also, what are SMO, IBk, J48, JRip in section 2.4? are they specific types of classifiers? They should be spelled out with providing a reference.
We thank the reviewer for this valuable comment. We added references to all ML methods and improved descriptions in section 2.4.
Reviewer 2 Report
Overall
The very interesting and relevant study presented in the manuscript proposes a machine learning-based methodology to reveal and estimate glioma grade-specific co-enrichment mechanisms and also correct for the batch effects found in the TCGA and UCSC Xena datasets. Basically, using interpretable machine learning the interactions among biomarkers were extracted and validated using data from CGGA, MSigDB. The work is highly significant for the community due to its potential to improve therapies and the same time elucidate how new pathways can be used in therapies. Overall excellent and thorough work, but some polishing is needed to make it sound less like a lab report.
Comments
As you are considering the impact of the batch effect on datasets that may remove biologically-relevant information that drastically affects the statistical analysis, I would recommend the work on DeepPAMM: Deep Piecewise Exponential Additive Mixed Models for Complex Hazard Structures in Survival Analysis.
As you look also for interpretable predictions under batch effects I would suggest also looking into the importance-based sequential procedures that identify the stable and well-performing combinations of features in the grouped feature space. Furthermore, I would suggest looking at the combined features effect plot, which is a technique to visualize the effect of a group of features based on a sparse, interpretable linear combination of features.
I would suggest adding a set of rules defined in your RBM such that the reader can understand how you adjusted connection values on the network to normalize the co-enrichment that may occur due to the overlapping gene sets.
As you pointed out PCA could be used when highly confounded batch effects are present. Could you please elaborate, in your case would the global or the local structure of interactions be more valuable? I would suggest exploring also the use of t-SNE. Preserving only local similarities will only (almost) completely ignore the global structure. When the global structure is lost, some data points may have been blocked from moving to their neighbours, and separation between different groups is not preserved quantitatively (i.e. relevant relations lost?). PCA is quite the opposite. It tries to preserve the global properties (eigenvectors with high variance) while it may lose low-variance deviations between neighbours (i.e. redundancy lost?).
I would add a bit more details on how your targeted investigation would have the potential of becoming a "methodology to perform ML analysis on less noisy data that may reveal potential interactions among pathways."
The manuscript reads now more like a lab report, with too many technicalities in the main body of the work obscuring the results and insights you gained. Please polish the sections by sending technicalities in the appendix or supplemental. For instance, if one doesn't use R, then your statement on line 353 "... applied multiinformation on annotations discretized into three levels with the same method as ML, i.e., equal frequency discretization" is misdirecting the reader...". You could rewrite as "we calculated total correlation on annotations ...."
Author Response
Response to Reviewer 2 Comments
Overall
The very interesting and relevant study presented in the manuscript proposes a machine learning-based methodology to reveal and estimate glioma grade-specific co-enrichment mechanisms and also correct for the batch effects found in the TCGA and UCSC Xena datasets. Basically, using interpretable machine learning the interactions among biomarkers were extracted and validated using data from CGGA, MSigDB. The work is highly significant for the community due to its potential to improve therapies and the same time elucidate how new pathways can be used in therapies. Overall excellent and thorough work, but some polishing is needed to make it sound less like a lab report.
Comments
As you are considering the impact of the batch effect on datasets that may remove biologically-relevant information that drastically affects the statistical analysis, I would recommend the work on DeepPAMM: Deep Piecewise Exponential Additive Mixed Models for Complex Hazard Structures in Survival Analysis.
We highly appreciate the reviewer's suggestion. However, the survival analysis was not our main goal, and we used basic solutions to provide general overview of the survival. We elaborated on this issue in the discussion and provided references to DeepPAM suggesting that this may be a more comprehensive way of assessing survival. As the DeepPAM seems not to be publicly available yet, we provided the reference manually based on https://www.slds.stat.uni-muenchen.de/research/#2022. In addition, we included a reference to other work describing DeepPAM that is public on arxiv.
As you look also for interpretable predictions under batch effects I would suggest also looking into the importance-based sequential procedures that identify the stable and well-performing combinations of features in the grouped feature space. Furthermore, I would suggest looking at the combined features effect plot, which is a technique to visualize the effect of a group of features based on a sparse, interpretable linear combination of features.
In contrast to importance-based sequential procedures, we focused on non-linear dependencies that are specific for rough sets and Monte Carlo feature selection (MCFS). We specified in the text that rough sets and MCFS aim at finding non-linear combinations of features. Specifically, MCFS finds combinations of features using decision trees and R.ROSETTA finds local combinations, i.e. sets of minimal combinations of features that preserve discernibility of the original set of features.
I would suggest adding a set of rules defined in your RBM such that the reader can understand how you adjusted connection values on the network to normalize the co-enrichment that may occur due to the overlapping gene sets.
We thank the reviewer for this valuable suggestion. We provided sets of significant rules for CGP and GOCC RBMs as Tables S6 and S7. We updated the section 3.4 “Glioma co-enrichment” and the caption of Figure 4.
As you pointed out PCA could be used when highly confounded batch effects are present. Could you please elaborate, in your case would the global or the local structure of interactions be more valuable? I would suggest exploring also the use of t-SNE. Preserving only local similarities will only (almost) completely ignore the global structure. When the global structure is lost, some data points may have been blocked from moving to their neighbours, and separation between different groups is not preserved quantitatively (i.e. relevant relations lost?). PCA is quite the opposite. It tries to preserve the global properties (eigenvectors with high variance) while it may lose low-variance deviations between neighbours (i.e. redundancy lost?).
In our case, a global structure would be more valuable as ML requires a sufficient number of samples, i.e. as many as possible. We explored t-SNE plots for raw data and attached it to the Supplementary materials (Figure S4). We elaborated the differences between local and global structure in the discussion.
I would add a bit more details on how your targeted investigation would have the potential of becoming a "methodology to perform ML analysis on less noisy data that may reveal potential interactions among pathways."
We thank the reviewer for pointing that out. We clarified this sentence in the text.
The manuscript reads now more like a lab report, with too many technicalities in the main body of the work obscuring the results and insights you gained. Please polish the sections by sending technicalities in the appendix or supplemental. For instance, if one doesn't use R, then your statement on line 353 "... applied multiinformation on annotations discretized into three levels with the same method as ML, i.e., equal frequency discretization" is misdirecting the reader...". You could rewrite as "we calculated total correlation on annotations ...."
We polished several sections in order to remove technical aspects about methods from the text. To this end, we have provided an additional Table S1 where we included all the R packages and functions used in this work. We also rewrote the text accordingly.